

# Tunable theranostics: innovative strategies in combating oral cancer

Asmaa Uthman[1], Noor AL-Rawi[2], Musab Hamed Saeed[3,4], Bassem Eid[5] and Natheer H. Al-Rawi[6,7]

[1] Department of Diagnostic and Surgical Dental Sciences, College of Dentistry, Gulf Medical University, Ajman, United Arab Emirates
[2] Department of Pharmaceutics and Pharmaceutical Technology, University of Sharjah, Sharjah, United Arab Emirates
[3] Department of Clinical Sciences, College of Dentistry, Ajman University, Ajman, United Arab Emirates
[4] Ajman University, Centre of Medical and Bio-allied Health Sciences Research,, Ajman, United Arab Emirates
[5] Department of Restorative Dental Sciences, College of Dentistry, Gulf Medical University, Ajman, Ajman, United Arab Emirates
[6] University of Sharjah, Sharjah Institute of Medical Research, Sharjah, United Arab Emirates
[7] Department of Oral and Craniofacial Health Sciences, College of Dental Medicine, University of Sharjah, Sharjah, United Arab Emirates

## ABSTRACT

**Objective.** This study aims to assess and compare the potential of advanced nano/micro delivery systems, including quantum dots, carbon nanotubes, magnetic nanoparticles, dendrimers, and microneedles, as theranostic platforms for oral cancer. Furthermore, we seek to evaluate their respective advantages and disadvantages over the past decade.

**Materials and Methods.** A comprehensive literature search was performed using Google Scholar and PubMed, with a focus on articles published between 2013 and 2023. Search queries included the specific advanced delivery system as the primary term, followed by oral cancer as the secondary term (*e.g.*, "quantum dots AND oral cancer," *etc.*).

**Results.** The advanced delivery platforms exhibited notable diagnostic and therapeutic advantages when compared to conventional techniques or control groups. These benefits encompassed improved tumor detection and visualization, enhanced precision in targeting tumors with reduced harm to neighboring tissues, and improved drug solubility and distribution, leading to enhanced drug absorption and tumor uptake.

**Conclusion.** The findings suggest that advanced nano/micro delivery platforms hold promise for addressing numerous challenges associated with chemotherapy. By enabling precise targeting of cancerous cells, these platforms have the potential to mitigate adverse effects on surrounding healthy tissues, thus encouraging the development of innovative diagnostic and therapeutic strategies for oral cancer.

## INTRODUCTION

Oral cancer is the most common violent cancer that infiltrates local tissue, and can metastasize to other body parts (*Ghazawi et al., 2020*). It is a challenging ailment affecting

Corresponding author
Natheer H. Al-Rawi,
nhabdulla@sharjah.ac.ae

approximately 600,000 people worldwide annually with a high rate of morbidity and mortality (*Ahmadian et al., 2019*). Despite the extensive research on this cancer over the past 20 years, the survival rate has not considerably increased (*Chang et al., 2013*), which is estimated to be about 50 to 60%, with a wide range recurrence rate of 18–76% (*Ahmadian et al., 2019*). The conventional diagnostic approach involving biopsy or histological assessment not only consumes valuable time but also delays the initiation of treatment, thereby further diminishing the prospects of favorable outcomes (*Hasanzadeh, Shadjou & Guardia, 2017*). Consequently ,the early detection of oral cancer emerges as a pivotal determinant for enhancing patient's survival prospects (*Poonia et al., 2017*). The standard treatment modalities for oral cancer encompass surgery, radiotherapy, and chemotherapy (*Kakabadze et al., 2020*). However, the drugs employed in conventional chemotherapeutics regimens suffer from deficiencies in term of systemic stability, water solubility, and tolerability, culminating in undesirable toxicities. These limitations, including a limited half-life and associated side effects such as bone marrow depression and nephrotoxicity, pose significant constraints on their future clinical application (*Wang et al., 2015*). Due to the tumor's specific anatomy and pathophysiological conditions, such as angiogenesis, hypoxia, low extracellular pH, and lack of lymphatic drainage, it is feasible to surmount conventional limitations of diagnosis and therapy with various nanoparticles (NPs) (*Jain & Stylianopoulos, 2010*).

Cancer nanotechnology is the most recent advancement in the field of cancer therapy. According to *Dhar & Shivji (2021)* the utilization of nanotechnology exhibits significant potential in enhancing cancer treatments through two primary mechanisms. Firstly, it enables the enhancement of pharmaceutical agents by improving their stability, modifying their pharmacokinetics, and reducing their toxicity. Secondly, nanotechnology facilitates the targeted delivery of drugs directly to the tumor site (*Dhar & Shivji, 2021*).

The utilization of nanomedicine and combinatorial chemotherapy has emerged as a promising therapeutic approach to address the limitations of existing treatment methods, which are hindered by the unintended adverse effects inflicted on healthy cells and the development of resistance in tumor cells (*Ren et al., 2020*). Nanotechnology refers to the deliberate manipulation of materials at the molecular and atomic levels. Nanomaterials are classified as substances that possess components with dimensions less than 100 nm in at least one direction. According to *Bhardwaj et al. (2014)*, the properties of nanoparticles, such as their electrical, optical, and magnetic characteristics, undergo transformation due to their diminutive size. Cancer detection methods based on nanotechnology permit the visualization of cytologic and morphological aberrations of malignant cells, such as alterations in nuclear size, epithelial thickness, and blood flow. Additionally, it can distinguish between premalignant lesions and malignancies. Surgical benefits are associated with nanotechnology–based detection methods, which aid in the identification of lymphatic metastasis and tumor margins (*Nassir, 2022*).

The synthesis of nanomaterials is often conducted using one of the two approaches: top-down and bottom-up methodologies. The top-down approach entails the fragmentation of macroscopic bulk materials into nanoparticles of significantly smaller dimensions. The utilization of top-down approach is characterized by its ease of implementation.

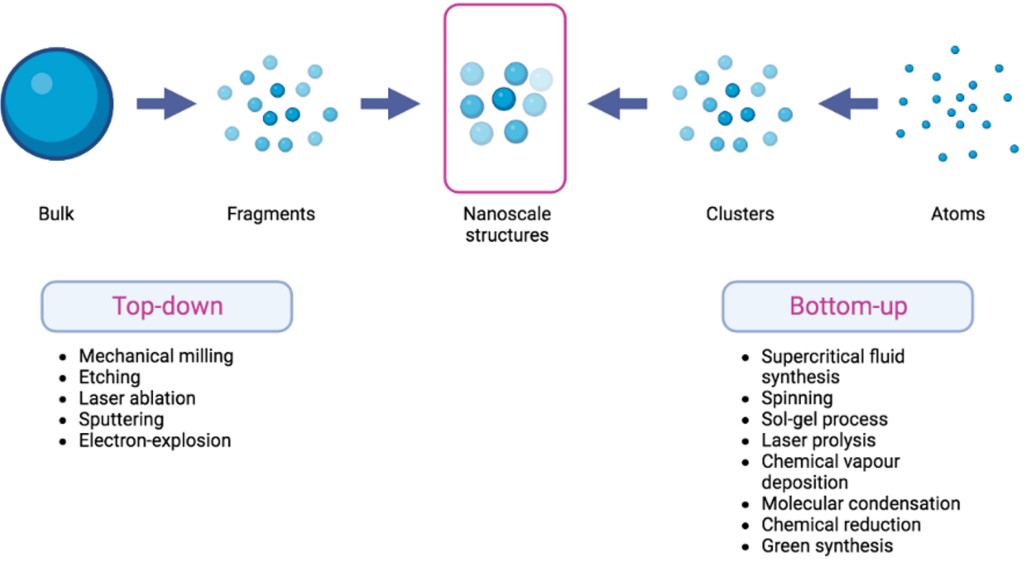

**Figure 1** Scheme of top-down and bottom-up methods in the fabrication of nanoscale structures.

Nevertheless, a significant limitation of this particular strategy lies in the challenge associated with achieving accurate particle size and shape (*Abid et al., 2022*).

The constructive technique, often known as the bottom-up strategy, exhibits a stark difference when compared to the top-down approach. Atoms and molecules play a fundamental role in the composition of nanomaterials as they undergo growth and self-assembly processes to generate nanoparticles that possess distinct characteristics in terms of their structure, size, and chemical composition (*Abid et al., 2022*). Figure 1 depicts the top-down and bottom-up approaches , accompanied by corresponding examples for each methodology. The concept of "theranostics" refers to the integration of diagnostic biomarkers and therapeutic medications that share a common target within afflicted cells or tissues (*Moorthy & Govindaraju, 2021*). Nanocarrier systems have the potential to be utilized in conjunction with laser-assisted therapy, photodynamic therapy (PDT), photothermal therapy (PTT), and/or imaging modalities (*Tade & Patil, 2020*). The enhanced permeability and retention (EPR) effect pertains to the capacity of macromolecules to selectively collect within the interstitial space of a tumor, in conjunction with a substantial influx of blood plasma and subsequent gradual clearance (*Nakamura et al., 2016*). The present focus in the creation of nanoparticles (NPs) mostly revolves around their application in the diagnosis and treatment of cancer, using the (EPR) effect and passive targeting strategies. Considering the inherent variability of the (EPR) impact inside the tumor microenvironment, there is an urgent need for the development of active targeting techniques that are both more accurate and efficacious. Hence, the utilization of NPS can involve its conjugation with ligands or antibodies for the purpose of detecting

tumor-specific receptors, such as anti-epidermal growth factor receptor (anti-EGFR) (*El-Sayed, Huang & El-Sayed, 2005*).

The nano systems, including quantum dots (QDs), carbon nanotubes (CNTs), magnetic nanoparticles (MNPs), dendrimers, and microneedles (MNs) have emerged as prominent advancements in the field of cancer theranostics (*Poonia et al., 2017*). The present investigation looks at and compares advanced nano delivery methods, exploring their potential as theranostics platforms, and evaluating their current developments and limitations in the context of oral cancer. A literature search was conducted using Google Scholar and Pubmed, focusing on articles published between 2013 and 2023. The search terms used were the specific advanced delivery system as the primary phrase like the quantum dots (QDs), carbon nanotubes (CNTs), magnetic nanoparticles (MNPs), dendrimers and microneedles (MNs) followed by oral cancer as the secondary phrase. Tables 1 and 2 summarizes the comparison between the advanced nano/ micro formulations as theranostics platforms in combating oral cancer.

## Quantum dots

Quantum dots (QDs), which are semiconductor nanocrystals, possess a quasi-zero-dimensional structure and exhibit robust photoluminescence (PL) without the need for surface functionalization. According to *Iannazzo, Celesti & Espro (2021)* these materials exhibit distinctive electrical characteristics, together with inherent fluorescence features that demonstrate exceptional resistance to photochemical degradation and photobleaching. The majority of QD synthesis methods exhibit complexity, high cost, time-consuming, and necessitate the utilization of toxic organic solvents, hence requiring surface passivation. The basic structure of QDs encompasses a semiconductor core, shell, and a cap. The shell serves as a protective covering, and can incorporate many targeting ligands such as peptides, proteins, antibodies, nucleic acids, aptamers, small molecules, or other chemical moieties. These ligands are capable of binding to specific target antigens. On the other hand, the cap improves solubility in aqueous fluids. QDs have an optical property that makes them emit visible light with various wavelengths when exposed to ultraviolet (UV) light. The wavelength of light emitted by QDs depends on their size. When subjecting smaller quantum dots (QDs) to UV light, they exhibit the emission of visible light with higher energy levels, often in the blue spectrum. When larger quantum dots are exposed to UV light, they produce red visible light with reduced energy. The emission of light is also dependent upon the constituent materials of the core of quantum dots (*Dhanabalan et al., 2017*). It is possible to separate QDs fluorescence signal from background auto-fluorescence, which improves detection sensitivity. In contrast to organic dyes, QDs exhibit a significantly extended fluorescence lifetime typically ranging from 10 to 40 nanoseconds. Additionally, QDs possess a notably higher brightness, being approximately 10 to 20 times brighter than individual molecules of organic fluorophores (*Badıllı et al., 2020*).

QDs have a high surface area to volume ratio, enabling them to provide numerous locations for the attachment of therapeutic agents, such as anticancer drugs, as well as multiple diagnostic components, including optical, magnetic, and radioisotopic moieties. According to *Tade & Patil (2020)*, it is possible to attach approximately 50 tiny molecules

**Table 1 Comparison between the advanced nano/ micro formulations as theranostics platforms.**

| | Particle size | General features | Disadvantages | Advantages in oral cancer theranostics | Limitations in oral cancer theranostics |
|---|---|---|---|---|---|
| Quantum dots (QDs) | 2–100 nm | ●Broad absorption spectra ● Narrow emission spectra from visible to near-infrared wavelengths ● Long lasting light ● High brightness ● Micro-range detection ● Delivers drugs and nucleic acids. | ● Leaching free metal ions ● Size <2.5 nm can interact with the lung alveolar ● Environmental concerns ● Complex, expensive synthesis ● Toxic solvents | ●Exceptional fluorescence properties for cancer detection ● Potential for photothermal therapy. | ●Potential toxicity ● Complexity of their synthesis & functionalization |
| Carbon nanotubes (CNT) | Diameter 1–2 nm for SWCNT. 100 nm for MWCNT | ●Mechanical resistance ● Light weight ● Electronic properties ● Optical properties ● Act as contrast agents ● Drug and gene delivery | ● Requires purification. ● Formulation challenges due to CNT's high hydrophobicity ● Increases dispersion viscosity. ● Toxicity issues may require CNT entrapment in nanocomposites. ● If immunity doesn't recognize CNT, they may accumulate in organs indefinitely | ●CNTs can be functionalized with various molecules, such as targeting ligands or imaging agents, allowing for targeted delivery, and imaging of oral cancer cells or tumors. ● CNTs can be used as sensors to detect specific biomarkers associated with oral cancer, enabling early detection and monitoring of the disease | ●Induce cytotoxicity and inflammation. ● High-quality CNTs with consistent properties can be complex and costly |
| Magnetic Nanoparticles | 10–100 nm | ●Multifunctional and stimuli-responsive ● High surface-to-volume ratio ● Used for hyperthermia, photothermal, and photodynamic therapies. ● Used for MRI, Magnetic Particle Imaging (MPI), and multimodal imaging. ● Allows for targeted, sustained, and pulsatile release. ● Delivers drugs and nucleic acids. | ● Low biocompatibility ● Insufficient magnetic strength ● Low drug loading capacity ● Difficult to control their sizes. | ●Precise targeting and localization of the nanoparticles to the tumor site, enhancing the specificity and accuracy of diagnosis and treatment. ● Used in MRI to provide detailed anatomical and functional information about oral cancer. ● MNPs can be utilized for hyperthermia therapy to selectively destroy cancer cells. | ●Potential toxicity of MNPs, particularly when they are not properly coated or functionalized. ● Challenges in achieving optimal accumulation and retention at the tumor site. |
| Dendrimers | 1–100 nm | ●Pharmacokinetics and biodistribution improvement of drugs ● High structural homogeneity and reproducibility ● Improved drug solubility ● Photodynamic therapy ● Controlled and targeted drug and gene delivery ● Stable and can be used in prodrugs design. ● Imaging and radiocontrast agents | ● High cost of production ● Requires control tests to guarantee dendrimers safety and quality. ● Cationic dendrimers e.g., PAMAM are cytotoxic and accumulate in the liver | ●High loading capacity and ability to encapsulate various therapeutic agents, such as drugs, imaging agents, and nucleic acids. ● Precise control over the release of therapeutic agents, enabling sustained and controlled drug delivery to the tumor site. ● Siimultaneous imaging and therapy, providing real-time monitoring of the treatment response and facilitating personalized medicine approaches. | ●Potential toxicity, particularly when used at high concentrations or with certain surface modifications. ● Synthesis of dendrimers with precise control over their size, shape, and surface functionalities can be complex and time-consuming. |
| Microneedles | Height: 25 to 2,000 um. Diameter: a little more than 30 um | ●Painless and non-invasive delivery ● Simple and convenient self-administration without special skills ● High biosafety ● Can be used in biosensing and immunotherapy. ● Can be conjugated with photothermal or photodynamic therapies. ● Drug and nanocarriers delivery | ● Skin reactions/ allergies ● Infections due to microorganisms entering the microchannels created. ● Contact dermatitis ● Skin redness (erythema) ● Hyperpigmentation and inflammation related to ceramic/ glass MN ● Biocompatibility and toxicity related to silicon/ metal MN ● Edema, scaling, and stinging or prickling sensation. ● Bruising and scarring after long-term use. ● Limited dose-carrying capacity | ●Provide minimally invasive and painless drug delivery allowing for targeted delivery of therapeutic agents to oral cancer cells or tumors. ● MNs can be designed to encapsulate or coat various types of therapeutic agents, including drugs, imaging agents, and vaccines enabling simultaneous diagnosis and treatment of oral cancer. | ●Challenge of achieving precise and uniform penetration depths which may affect the efficacy and reliability of therapy. ● Fabrication process of microneedles can be complex and time-consuming, requiring specialized equipment and techniques. |

**Table 2** Advanced nano/ micro platforms in combating oral cancer.

| Type of NP | Anti-cancer agent/-dose | Dosage form | Synthetic procedure | Experimental setting/-duration | Cell lines or animal model | Main findings | Refs |
|---|---|---|---|---|---|---|---|
| Graphene QDs functionalized with FA | Evodiamine/ 200 μL | Tail vein injection | Self-assembly | *In vivo*/18 days | Tumor bearing BALB/c mice | *In vivo* whole-body fluorescent images displayed the growth inhibition rate of OSCC cells to exceed 50% ($p < 0.01$) when the composite system was loaded with 10% EVO | Ma et al. (2022) |
| QD800-EGFR Ab | Nil | Subcutaneous transplant | Conjugation | *In vitro*/24 hrs | Animal model | Fluorescence signals of BcaCD885 cells labeled with QD800-EGFR Ab probe could be clearly detected, and these fluorescence signals lasted for 24 h. | *Yang et al. (2011)* |
| QD800-RGD | Nil | Tail vein injection | Conjugation | *In vitro*/9–12 hrs | Animal model | QD800-RGD was specifically targeted to integrin αvβ3 *in vitro* and *in vivo*, producing clear tumor fluorescence images after the intravenous injection. | *Huang et al. (2013)* |
| Graphene QDs functionalized with PEG | GQD-PEG/ 100 μg/mL | Tail vein injection | Bottom up method | *In vitro*/24 hrs *In vivo*/24 hrs | Human OSCC cell lines SCC 25 and SCC 9 C3H murine bearing SCC VII tumor | After incubation with 100 μg/mL of GQD-PEG and light irradiation for 10 min, SCC 9 and SCC 25 cells showed decreases of 70% and 30%, respectively, in contrast to control groups The "GQD-PEG plus irradiation" group elicited tumor size reduction exceeding 70%, compared with those of the "laser only" and "PBS plus laser" groups | *Zhang et al. (2020)* |
| Vertically-aligned Carbon Nanotube Interdigitated Electrodes | Nil | Chip | Chemical Vapor Deposition (CVD) process | Clinical | 10 μL of human saliva supernatant | In saliva, the developed sensor has a maximum sensitivity with a linear range of 1–100 pg/mL and a detection limit of 0.24 pg/mL, compared to the CIP2A enzyme linked immunosorbent test (detection range: 0.156 ng–10 ng/mL) | *Ding et al. (2018)* |
| iron–gold bimetallic nanoparticles | Hyperthermia and MMP-1 | Thermal pyrolysis | Conjugation | *In vitro*/24 hrs | HSC-3 | MMP-1-FeAu NPs conjugate triggered 89% HSC-3 cellular death, confirming the efficacy of antibody-conjugated nanoparticles in limiting SCC growth. | *Tsai et al. (2021)* |

**Table 2** (*continued*)

| Type of NP | Anti-cancer agent/-dose | Dosage form | Synthetic procedure | Experimental setting/-duration | Cell lines or animal model | Main findings | Refs |
|---|---|---|---|---|---|---|---|
| Superparamagnetic iron oxide (SPIO) | Nil | Magnetic resonance lymphography | Contrast enhanced medium | *In vivo* | Clinical study | All SLNs could be detected 2 min and 3.5–5 min after contrast medium injection. In all patients, SLNs were detected by MRL at 10 min after SPIO injection, and the total and mean number of SLN was 53 and 2.7, respectively. MRL at 30 min after the injection showed additional 18 secondary lymph nodes. | Sugaiyama et al. (2021) |
| Iron oxide Magnetic Nanoparticles (MNPs) | Hyperthermia and $\alpha v\beta6$ antibody/ 10 μg/ml | Dispersion | Co-precipitation method | *In vitro*/ 24–48 hrs | H357 and Vβ6 cell monolayers | *In vitro* cytotoxicity experiments on Vβ6 cells demonstrated cell survival below 50% in the first 24 hr and below 25% in 48 hr. Hyperthermia decreased cell survival after 24 and 48 h. For H357, heat only affected cell survival after 48 h. | *Legge et al. (2019)* |
| Iron oxide MNPs | Nil | Dispersion | Oxidative hydrolysis | *In vitro* | Ca9-22 and CAL 27 | MNP-delivered siBCL2 reduced BCL2 mRNA by 18% in Ca9-22 and 56% in CAL27 cells. Western blot tested for protein. BIRC5 silencing was similar. MNP+siBCL2 and MNP+siBIRC5 reduced tumor cell viability. | *Jin et al. (2019)* |
| PMAM dendrimer | Methotrexate | Injection | Acetylation | *In vivo*/10–15 days | Mice | Targeted methotrexate treatment was investigated in vivo on 3 cell-line xenografts with varied folate receptor expression (no, moderate, and high). Treatment of high folate expression tumor cells showed higher effectiveness ($P < .01$) and less systemic toxicity ($P = .03$) compared to saline and free methotrexate. | *Ward et al. (2011)* |
| Linear-dendritic mPEG-BMA4 | Saracatinib/1.0 mg | Injection | Solvent evaporation method | *In vivo*/48 h | NSG mice | Saracatinib was able to suppress Src-dependent invasion/metastasis by reducing Vimentin and Snail protein expression. Nanosar was also favorable to saracatinib in its ability to prevent tumor metastasis | *Lang et al. (2018)* |
| PMAM dendrimer | FITC-G4-FA) | Gene therapy | Conjugation | *In vitro* | HN12-YFP and U87 cells | G4-FA seems most efficient in transfecting HN12 cells and maintains good cytocompatibility when it is complexed with plasmid at a weight ratio of 5:1. | *Xu et al. (2016)* |

**Table 2** (*continued*)

| Type of NP | Anti-cancer agent/-dose | Dosage form | Synthetic procedure | Experimental setting/-duration | Cell lines or animal model | Main findings | Refs |
|---|---|---|---|---|---|---|---|
| PLGA NPs encapsulating DOX were coated on microneedles | Doxorubicin/8μl | Microneedles | Double emulsion w/o/w and wet itching process | *In vitro* | Porcine cadaver buccal tissue | DOX-PLGA-NP/MNs showed that DOX could dissipate up to 4 mm deep into porcine cadaver buccal tissues, causing cell death. Coated microneedles perform better than hypodermic needles for they can deliver drugs at precise and evenly distributed tissue locations while avoid drug loss and systemic side effects from injection fluid leakage. | *Ma et al. (2015)* |
| αCTLA-4 MN patches | anti-CTLA-4 | Microneedles | Molding | *In vivo* | Mouse | local IT delivery of αCTLA-4 ICI will initiate a robust and durable antitumor response that is dependent on cDC1 and CD8+ T-cells, while significantly limiting irAEs. | *Gilardi et al. (2022)* |

onto quantum dots (QDs) with a diameter of 4 nm using various mechanisms such as covalent bonding, electrostatic forces, multivalent chelation, or passive adsorption. Inorganic QDs are composed of metal complexes such as CdSe, ZnS, CdTe that are primarily accompanied by an organic coating to augment their biocompatibility or bioactivity. The potential toxic effects resulting from contact with the inorganic core may manifest following the breakdown of the organic coating. Therefore, the study of graphene quantum dots (QDs) without the presence of heavy metal elements has been conducted with the aim of applications in imaging, drug delivery, gene transfer, and cancer therapy (*Chaturvedi et al., 2019*). The excellent optical properties of quantum dots (QDs) have been utilized extensively in the diagnosis of cancer cells. For example, *Yang et al. (2011)* have fabricated EGFR-antibody-conjugated QD800 (QDs with a maximum emission wavelength of 800 nm) for the targeting and *in vivo* imaging of human buccal squamous cell carcinoma cell line (BcaCD885) in an OSCC animal model, and QD800 has strong tissue penetration, making it suitable for visible fluorescence imaging (*Yang et al., 2011*). Currently, the majority of QDs used for *in vivo* imaging are targeted to antigens or receptors that are highly expressed on cancer cell surfaces or specifically expressed in tumor cells; QDs coupled with respective antibodies or ligands can bind specifically to these targets (*Diagaradjane et al., 2008*). Although fluorescence probes can reach tumor cell surfaces *in vivo*, their efficacy may be diminished since they must first pass through the tumor capsules and vascular walls. Recently, quantum dots with near-infrared emission spectra have emerged as essential tools for *in vivo* tumor imaging due to their exceptional tissue penetration and photostability. Integrin v3 binds specifically to the peptide containing arginine-glycine-aspartic acid (RGD) and has been shown to be highly and specifically expressed in endothelial cells of tumor angiogenic vessels in nearly all forms of tumors. In the study by *Huang et al. (2013)* RGD was conjugated with quantum dots with an emission

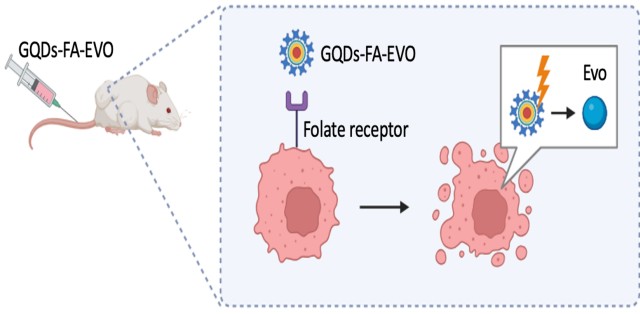

**Figure 2** **GQDs-FA-EVO self-assembly fabrication process, targeting and drug treatment, and tumor labeling.** Adopted from *Ma et al. (2022)*.

wavelength of 800 nm (QD800) to create QD800-RGD, which was injected intravenously to image tumors in nude mice with head and neck squamous cell carcinoma (HNSCC). The results demonstrated that QD800-RGD targeted integrin v3 specifically *in vitro* and *in vivo*, producing clear tumor fluorescence images after intravenous administration.

In relation to the therapeutic effectiveness of quantum dots (QDs), in a recent investigation conducted by *Ma et al. (2022)*, Graphene Quantum Dots (GQDs) that were functionalized with folic acid (FA) were employed as a targeted vehicle for the administration of evodiamine (EVO). This bioactive compound has been demonstrated to effectively inhibit the proliferation of oral squamous cell carcinoma (OSCC), as depicted in Fig. 2. Results indicated the formation of a highly stable nanocomposite complex, with improved water solubility, high biocompatibility, as well as reduced toxicity *in vitro* and in vivo. Furthermore, following 24 hr of injecting GQDs into the mice, the tumor localization patterns were visualized by *in vivo* whole body florescent imaging at 480 nm excitation wavelength. GQDs loaded with 10% EVO led to a growth inhibition rate of OSCC cells exceeding 50% ($p < 0.01$). The tumor volume in tumor-bearing nude mice decreased by 19% with GQDs-FA-EVO composite, compared to the EVO group ($p < 0.05$) during the 18-days treatment protocol. In another effort, *Zhang et al. (2020)* used GQDs as photosensitizers for photodynamic therapy and immunostimulatory activity, conjugated to polyethylene glycol (PEG) to enhance solubility and blood circulation as depicted in Fig. 3.

The fluorescence signal intensity was assessed *in vitro* and *in vivo* to determine phototoxicity and tumor uptake in an OSCC mouse model after irradiation. GQD-PEG showed no cytotoxicity, high solution stability, and excellent endocytosis. The substance's photodynamic efficacy was also strong *in vitro* and *in vivo* (*Zhang et al., 2020*).

## Carbon nanotubes

Carbon nanomaterials integrate exceptional characteristics such as a very strong chemical resistance (not dissolving even in harsh settings), great mechanical qualities and a very low weight. The most widely used carbon nanomaterials involve carbon nanotubes (CNT), nanodiamonds (ND), additionally to graphene and its related materials (GRM) (*Simon, Flahaut & Golzio, 2019*). Carbon nanomaterials showcase a span of morphologies from
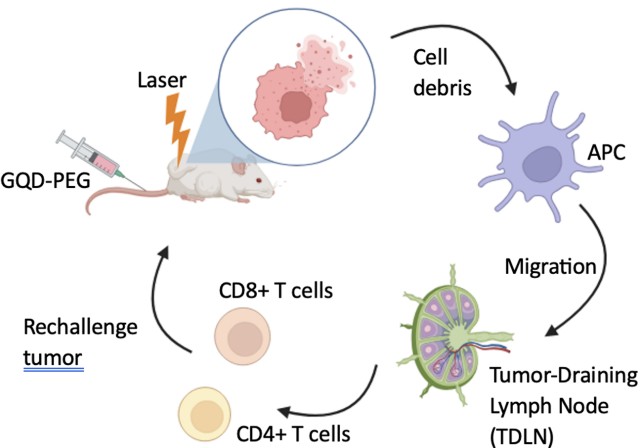

**Figure 3 Schematic illustration of GQD-PEG-mediated photo-triggered immune responses for tumor therapy.** Adopted from *Zhang et al. (2020)*.

0D (nanodiamonds) to 1D (carbon nanotubes/nanowires) and 2D (GRM nanosheets or nanoplatelets). Among them, CNT showcase a rare mix of mechanical, optical, and electrical properties. Additionally, the capability to functionalize their interface with a variety of bio/chemical species opens up a wealth of therapeutic and drug delivery opportunities (*Singh & Deshmukh, 2022*). CNT are axially concordant nanoscale tubular materials. They are described as a rolled- up graphene layer, occasionally closed at the end by fullerene caps (*Klochkov et al., 2021*). They can be created using laser ablation, Catalytic Chemical Vapor Deposition (CCVD), and electric- arc discharge processes. CNT may have a single wall (SWCNT) with a small diameter, typically 1 to 2 nm, or a multi-wall (MWCNT), with an outer diameter reaching 100 nm. MWCNT with many layers are easier to manipulate and functionalize, usually at the expense of losing some of their physical features. Double-wall CNT (DWCNT) are at the interface between SWCNT and MWCNT.

Single-walled carbon nanotubes (SWCNTs) have a very small diameter and excellent mechanical properties. Due to an exterior wall, they can be covalently functionalized without affecting their electrical conductivity. Mechanical properties of carbon nanotubes (CNTs) include a large surface area, up to 1,000 $m^2$ $g^{-1}$ for double-walled CNTs (*Simon, Flahaut & Golzio, 2019*). The electrical characteristics of the material facilitate rapid electron transfer and augment the sensitivity for electrochemical detection. In this regard, researchers have contemplated the utilization of carbon nanotubes (CNTs) as the principal constituent of electrochemical sensors and label-free biosensors based on CNTs. Additionally, the fluorescence of single-walled carbon nanotubes (SWCNTs) is emerging as a reliable method for optical imaging in the field of vascular observation (*Hendler-Neumark & Bisker, 2019*). Moreover, SWCNT fluorescence has been recently employed as an optical alternative to the costly positron emission tomography-computed tomography (PET-CT). Carbon nanotubes (CNTs) are currently the subject of intensive investigation in the fields of cancer treatment, medicine administration, and gene delivery (*Yudasaka et al., 2017*). Considerable attention has been directed towards the vertically aligned carbon

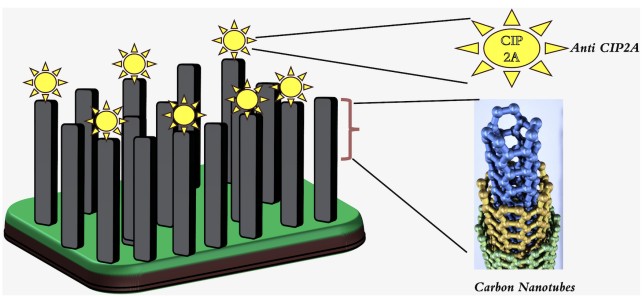

**Figure 4** **Schematic diagram showing anti-CIP2A Antibody bound to the VANTA IDE with CIP2A Antigen.** Adopted from *Ding et al. (2018)*.

nanotube array (VANTA) coating owing to its notable attributes of electrical conductivity, chemical inertness, and light absorption. It is conceivable to employ carbon nanotubes as immunosensors in the context of an oral cancer screening test. A 2D interdigitated electrode (IDE) footprint is utilized to deposit arrays of 3D high-aspect-ratio vertically aligned carbon nanotubes *via* chemical vapor deposition. The sensitivity of the biosensor was found to be higher than that of the ELISA test kit, as it detected the malignant inhibitor PP2A (CIP2A) in saliva supernatant without requiring preconcentration or pre-labeling of the sample as depicted in Fig. 4. This biosensor is also capable of reducing the total sensing duration (*Ding et al., 2018*).

## Magnetic nanoparticles

Similar to other nanomaterials, magnetic nanoparticles (MPs) are increasingly being utilized in the biomedical domain due to their functional surfaces and biocompatibility. Metallic nanoparticles, including iron oxide and gold, have the potential to function as X-ray contrast imaging agents owing to their X-ray absorption capabilities and short-term low toxicity (*Nune et al., 2009*). MNPs are classified as multifunctional nanomaterials as depicted in Fig. 5. Their dimensions range from 10 to 100 nm. Generally they consist of pure metals (Fe, Co, Ni, and other rare earth metals) or a a polymer-metal composite. In order to enhance the solubility of functional ligands, a coating that is both biocompatible and secure is applied (*Farzin et al., 2020*). MNPs are useful for targeted and controlled drug administration because they may be regulated by an external magnetic field, which also changes the pharmacokinetics of the drug and lengthens its half-life and release time (*Al-Rawi et al., 2020*). They are generally synthesized by coprecipitation, micro emulsion, and high temperature-based techniques, among many other techniques (*Yazdani & Seddigh, 2016*). MNPs are considered next-generation MRI contrast materials because they can be localized into the tissue locations to enhance proton relaxation and to improve their visibility. They can also increase the sensitivity of diagnostic equipment and biosensors (*Al-Rawi et al., 2020*). In addition, MNPs have large surface-area-to-volume ratios and high magnetic moments, rendering them highly attractive for the application of hyperthermia

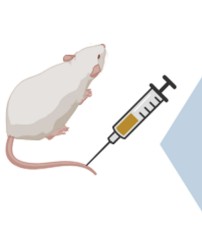

**Diagnostics**

- Computed tomography (CT)
- Magnetic Resonance Imaging (MRI)
- Single-photon emission computed tomography (SPECT)
- Photoacoustic Tomography (PAT)
- Florescence Imaging

**Therapy**

- Magnetic Hyperthermia (MHT)
- Photothermal therapy (PTT)
- Radio-chemotherapy

**Figure 5   Multifunctional magnetic-based theranostics nanoparticles.**

treatment in cancer therapy. Healthy cells can endure hyperthermia temperatures of 42–45 °C for a short duration (*Hegyi, Szigeti & Szász, 2013*).

In contrast, cancerous cells undergo apoptosis due to the reduction of pH within the cancerous microenvironment, which results in decreased thermotolerance (*Berthier et al., 2017*). Moreover, A disorganized vascular network and decreased blood flow within the malignant tissue slows the tumor's rate of convective cooling, leading to overheating. Radiofrequency, ultrasonic waves, infrared radiation, microwaves, and hot water can all stimulate MNPs to generate heat in a conventional hyperthermia treatment (*Zhao et al., 2020*). Hyperthermia therapies employ MNPs composed of the following elements: Fe, Mn, Zn, Mg, Co, Ni, Gd, and their oxides. Metal oxides are more biocompatible and stable *in vivo* than metals, they make better candidates for use in medical applications (*Farzin et al., 2020*). MNPs can be actively delivered to the tumor site to improve their localization. By employing this method, high-affinity ligands that binds to cell receptors adorn the surface of MNPs. Chemical modification offers multiplexed capabilities, including integrated drug delivery and hyperthermia and multimodal imaging (*Al Rawi et al., 2021*). Furthermore, chemotherapy and/or radiotherapy can be combined with hyperthermia to provide the following synergistic effects: (a) Most cells enter the synthesis phase (S phase) after being exposed to a certain dosage of ionizing radiation. Simultaneous exposure of hypoxic cells to hyperthermia therapy hinders DNA damage repair and boosts treatment effectiveness. (b) Hyperthermia improves the drug's rate of uptake and buildup in malignant tissue, which increases cell sensitivity to chemotherapeutic agents (*Farzin et al., 2020*). In 2014, Candido et al. tested polyphosphate-coated maghemite nanoparticles (MNPs) on oral squamous cell carcinoma using human oral cancer cells (UM-SCC14A) incubated with MNPs at various concentrations. MTT, apoptosis, and transmission electron image analysis were performed. Transmission electron microscopy demonstrated that the intermediate dosage tested *in vitro* was not hazardous. Therefore, this MNPs concentration was used for *in vivo* experiments. DMBA carcinogen was applied to Syrian hamsters to induce oral tumors. Animals received MNP-induced magneto-hyperthermia. Histopathological and immunohistochemical showed that mice treated with MNPs and subjected to the alternating magnetic field in hyperthermia had considerable and time-dependent cancer

regression. The oral cancer tumor-bearing Syrian hamsters recovered 100% (12/12) seven days following magneto-hyperthermia with these polyphosphate-coated MNPs. Data suggests MNPs-mediated hyperthermia may be promising method to treat oral cancer (*Candido et al., 2014*). In 2021, *Tsai et al. (2021)* coupled MMP-1 antibodies to iron–gold bimetallic nanoparticles (FeAu NPs) to examine magnetic field-induced hyperthermia's potential to target and restrict Scc cell proliferation. Superparamagnetic FeAu NPs (4.32 ± 0.79 nm) with a saturation magnetization of 5.8 emu/g were shown to be able to deliver hyperthermia at 45 °C. SCC targeting was confirmed by conjugation with MMP-1 antibodies, which increased human tongue squamous cell carcinoma cell uptake by 3.07-fold over fibroblast cells and decreased cell survival by 5-fold. MMP-1-FeAu NPs combination caused 89% HSC-3 cellular death during magnetic stimulation, proving antibody-conjugated nanoparticles restrict SCC development. They concluded that Biomarker-specific antibodies and magnetic nanoparticle-induced hyperthermia may enable SCC targeting for better disease prognosis (*Tsai et al., 2021*). For sentinel lymph nodes (SLN) mapping prior to surgery in 20 patients with clinically N0 oral cancer, *Sugiyama et al. (2021)* found that a total of 18 patients (90%) had SLNs detected by CT-lymphography (CTL). After 2–5 min following contrast medium injection, all SLNs were detectable. Magnetic resonance lymphography (MRL) detection of SLNs occurred 10 min after superparamagnetic iron oxide (SPIO) injection in all patients. MRL revealed 18 additional secondary lymph nodes 30 min after the injection. The researchers reached the conclusion that the utilization of superparamagnetic iron oxide (SPIO) in conjunction with magnetic resonance lymphography (MRL) is a safe and effective method for identifying sentinel lymph nodes (SLNs) in patients with clinically N0 early oral cancer. Furthermore, they determined that the most appropriate period for conducting the imaging procedure is 10 min following the injection of SPIO. The investigation conducted by Leggie et al. focused on iron oxide MNPs that were coated with biocompatible silica. In addition to targeting integrin $\alpha v\beta 6$, a well-established biomarker for oral squamous cell carcinoma, with MNPs conjugated with antibodies, tumor ablation was achieved *via* magnetic hyperthermia operating under an alternating magnetic field. *In vitro* cytotoxicity assays on V$\beta 6$ cells revealed a significant decline in cell survival percentage of below 50% for the av$\beta 6$-MNP group, compared to non-functionalized MNPs in the first 24 hr, and below 25% in the 48 hr. Additionally, hyperthermia significantly decreased cell survival after 24 and 48 h. In contrast, the cytotoxicity tests on H357 cells displayed no significant difference related to targeted and untargeted treatments, and hyperthermia only significantly affected cell survival in the 48-hour period. The killing capability of OSCC by magnetic hyperthemia could considerably increased with meticulous antigen, antibody, and nanoparticle coupling (*Legge et al., 2019*). *Jin et al. (2019)* identified an additional significant indication in the application of iron oxide MNPs modified with polyethyleneimine (PEI). These MNPs were designed to transport therapeutic siRNAs that specifically target B-cell lymphoma-2 (BCL2) and Baculoviral inhibitor of apoptosis repeat-containing 5 (BIRC5) into Ca9-22 oral cancer cells. With the help of quantitative real-time Polymerase Chain Reaction (PCR) and western blotting, high gene silencing efficiencies were identified. The mRNA level of BCL2 was significantly reduced by MNP-delivered siBCL2 to 18% in Ca9-22 cells and 56%
in CAL27 cells, compared with the MNP negative control groups. This was verified in the protein levels using western blotting. Similar outcomes for BIRC5 silencing were attained. Moreover, the anti-tumor activity was evaluated and a significant drop in cell viability was achieved by MNP+siBCL2 and MNP+siBIRC5 groups, compared with the negative group.

Su et al. (2019) investigated the use of anti-CD44 antibody-conjugated superparamagnetic iron oxide nanoparticles to target CD44, a well-characterized oral carcinoma biomarker that facilitates cancer cell immune evasion. The findings of this study indicate the feasibility of eliminating cancer stem cells (CSCs) by the utilization of specific magnetic nanoparticles in conjunction with an alternating magnetic field (AMF). Moreover, the application of magnetic fluid hyperthermia shown a substantial hindrance in the progression of Cal-27 tumors that were transplanted onto mice. In addition, thermochemotherapy can decrease the effective dosage of cisplatin. Sato et al., (2014) utilized ferucarbotran (commercial-grade super-paramagnetic iron oxide) in conjunction with cisplatin to suggest that combinations of magnetic hyperthermia and chemotherapy might be more effective than hyperthermia or chemotherapy alone. In order to enhance the efficacy and specificity of transgene therapy for oral squamous cell carcinoma (OSCC) in a xenograft model, Miao et al. (2014) developed a novel magnetic nanovector. In the presence of a magnetic field, positively charged polymer PEI-modified $Fe_3O_4$ magnetic nanoparticles were evaluated as gene transfer vectors. The $Fe_3O_4$ nanoparticles exhibited favorable solubility in water after being synthesized via co-precipitation. By electrostatic interaction, these PEI-modified nanoparticles were joined with negatively charged pACTERT-EGFP. In comparison to the frequently used PEI or lipofectin, the transfection efficiency of the magnetic complexes was as much as sixfold greater. In a xenograft model, they concluded that treatment with pACTERT-TRAIL delivered by magnetic nanoparticles induced apoptosis, which was a significant cytostatic effect. This suggests that employing magnetic nano-gene vectors with the plasmid pACTERT-TRAIL could enhance the efficiency of gene transfer for Tca83 cells and potentially induce antitumor effects. This could represent an innovative approach to managing OSCC (Miao et al., 2014).

## Dendrimers

Dendrimers are a group of globular macromolecules that are created from a core through. A repeated chemical process and have a predictable and highly organized structure. They are monodisperse and their sizes range between 1–100 nm (Moorthy & Govindaraju, 2021). The dendrimer structure is made up of three primary components: (1) the initiator core or nucleus, (2) the inner layers made of repetitive molecular units called dendrons, which originate radially from the core to form generations, and (3) the terminal groups on the surface resembling tree architecture (Dias et al., 2020). Dendrimers can be produced by different techniques, such as the divergent approach, in which the dendrimer develops from a core, which contains functional groups to react with monomers to produce the first generation of dendrimers (G1). The following step is the deprotection of the inactive groups on monomers, to enables the binding of more monomers and, eventually, dendrimer growth. Another approach is the convergent method, which starts from the dendrimer's exterior (monomer) to its inside (core). Other fabrication processes include

double exponential growth, "click" chemistry, and onion peel method (*Andreozzi et al., 2017*). Dendrimers can be used for RNA , DNA, and proteins delivery, as well as radiocontrast agents for imaging. They can hold bioactive compounds by forming covalent bonds with the surface (dendrimer prodrug), or by ionic interactions or adsorption in the interior of the nanostructure (*Mittal et al., 2021*). Moreover, dendrimers can be linked to targeting structures as antibodies, aptamers, carbohydrates, and folic acid, and are usually linked to other polymers, such as PEG (polyethylene glycol). Compared to linear polymers, dendrimers are favorable due to: (1) pharmacokinetics and biodistribution improvement of drugs due to their controllable size and shape (2) high chemical and structural uniformity, promoting pharmacokinetic reproducibility (3) the capacity to interact with numerous active substances and/or ligands, which can increase their solubility (4) targeted dendrimer design, which improves the specificity of nanocomplexes (5) control of dendrimer breakdown by chemical entities included in the structure (*Akbarzadeh et al., 2018*). The dendrimers extensively branched architecture offers significant adaptability for various modifications, particularly in the context of cell targeting, high-capacity drug encapsulation, gene therapy payloads, or their synergistic combinations (*Yuan et al., 2019*).

Most dendrimers utilized in cancer therapy are composed of Poly (amidoamine) (PAMAM). In cancer research, dendrimers are often employed in photodynamic therapy, a treatment that uses light and chemicals called photosensors that, produce reactive oxygen species when exposed to light with a specific wavelength that kill cancerous cells. Patients are intravenously administered with photosensors, and the compound accumulates in the tumor within 24–72 h. Then, the light is focused on tumor tissue, yielding reactive oxygen species to destroy the cells containing the photosensors (*de Araújo et al., 2018*). Another notable dendrimer therapeutic strategy is the employment of RNA interference (RNAi), that is responsible for genes silencing and acting on gene expression regulation, including microRNA (miRNA) and small interfering RNA (siRNA). The RNAi derivative dendrimer can attach to cell surfaces, get ingested (mainly by endocytosis), and then release the nucleotide. In the intracellular medium, RNAi may interact with the RNAi machinery to silence genes (*Castro, Forero-Doria & Guzman, 2018*). Dendrimers are extensively employed as highly promising options for delivery of drugs applications. In their study, *Ward et al. (2011)* conducted co- encapsulation of methotrexate, an anticancer medicine, and folic acid (FA), a targeting agent, within acetylated generation 5 dendrimer. The researchers observed improved tumor control compared to the use of free drug in xenograft tumor development models.

A set of DNA-dendrimer and polypyrrole (DDPpy) sensors have been devised by *Wei et al., (2009)* to identify oral cancer biomarkers, including interleukin-8 RNA, interleukin-8 protein, and interleukin-1β protein. These sensors demonstrate enhanced bio affinity and specificity. According to Lang and partners, dendritic nanoparticles (Nano-sar) loaded with saracatinib effectively inhibited the invasion and metastasis of head and neck squamous cell carcinoma (HNSCC) by inhibiting Src kinase activity, which is a non-receptor tyrosine kinase, that promotes the growth and metastasis of HNSCC tumors (Fig. 6). Using a mouse model of metastasis, saracatinib was able to suppress Src-dependent invasion/metastasis by reducing Vimentin and Snail protein expression. The superior effects over the free drug can

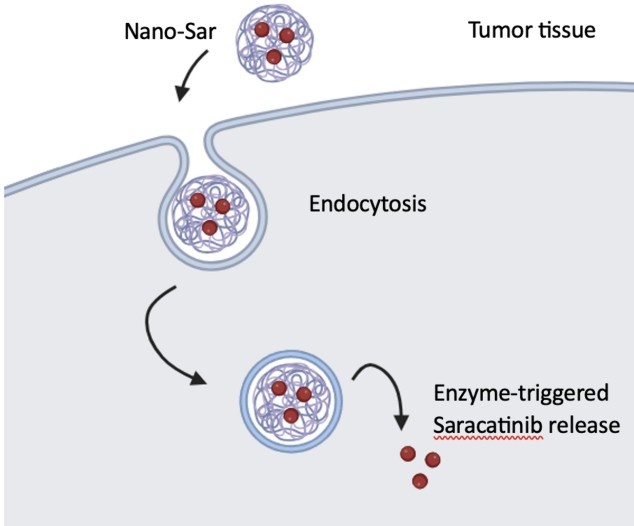

**Figure 6** **The production and operation of Nano-sar, (A) a schematic depiction of the self-assembling Nano-sar and its disintegration following cathepsin B (CTSB) digestion; (B) a schematic explanation of how Nano-sar targets tumor cells.** Adopted from *Lang et al. (2018)*.

be explained by its highly precise and efective tumor-specific drug delivery. Nano-sar was also favorable to saracatinib in its ability to prevent tumor metastasis (*Lang et al., 2018*).

A folic acid-modified polyamidoaminedendrimer G4 (G4-FA) nanoplatform was devised by *Xu et al. (2016)* to facilitate the precise transportation of DNA plasmids to head and neck cancer cells that exhibit high expression of folate receptors. G4-FA exhibits favorable cell compatibility and can vie for binding sites on cancer cells with free folic acid. For enhancing gene expression in cancer cells, G4-FA can bind specifically to folate receptors, thereby facilitating the incorporation of DNA plasmids. Furthermore, it can deliver the plasmids selectively to cancer cells that have a high expression of the folate receptor.

## Microneedles

MNs, as the name implies, are patches of micro-sized needles that are intended to deliver therapeutic drugs transcutaneously to the desired tumor site. They are a blend of hypodermic needles (except they are considerably smaller), and transdermal patches. MN range in height from 25 to 2,000 um and have an external diameter of a little more than 30 um. They are created utilizing various materials, including metal, polymer, silicon, ceramics, titanium, glass, and sugar-like carbohydrates (*Kwon et al., 2017*). MNs are categorized into the following groups according to the mode of drug delivery: solid, coated, porous, dissolving, and hydrogel-forming (*Al-Rawi & Rawas-Qalaji, 2022*). Solid MNs function by penetrating the skin and creating channels that allow the formulation to traverse the stratum corneum layer and then enter the capillaries to provide a systemic effect (*Hao et al., 2017*). In coated MN, the MNs are injected with a vaccine or drug suspension. The thickness of the coating solution affects how much medication can be placed into the MNs (*Yang et al., 2019*). Hollow MN have a hollow core and an aperture at the tip that allows

drug release, and controlling the drug flow produces rapid or sustained delivery (*Moreira et al., 2019*). Dissolving MNs typically dissolve in bodily fluids upon insertion and deliver the medication into the skin without a trace (*Al-Rawi & Rawas-Qalaji, 2022*). Hydrogel MN are fabricated by cross linking swellable polymers (hydrogels). When inserted within the skin, the hydrogel (hydrophilic in nature) absorbs water and swells. The advantage of this type of MN is the ability to deliver large doses without resulting in any drug loss (because the entire formulation is released within the skin). Its drawback on the other hand is the sustained release rate, which depends on the dissolution profile of the material used (*Ita, 2017*). To synthesize solid MN, techniques such as laser ablation, 3D printing, electroplating, dry or wet etching, and micro- molding are used. In contrast, coated MNs are created after solid MNs have been rolled, dipped, or sprayed. Hollow MNs can be made by micropipette pulling, isotropic or deep reactive ion etching, wet chemical etching, laser micromachining, or wet chemical etching. Finally, dissolvable MNs are fabricated by a method of negative molding (*Tucak et al., 2020*). The drug release profile may also differ according to the MN type, for instance, solid MN employ poke and patch method, while hollow MN use poke and flow method. Coated MN adopts coat and poke method, and lastly, dissolving MN utilizes poke and release method (*Al-Rawi & Rawas-Qalaji, 2022*). MNs have demonstrated desirable qualities, such as painless penetration, simplicity in self-administration, strong therapeutic benefits, and high biosafety. The versatile platform of MNs holds promise for medication delivery, as they are intended to transport both big and small medicinal molecules, heavy protein molecules, genes, antibodies, and nanoparticles (*Yang et al., 2022*). Furthermore, MNs have been frequently used in biosensing to achieve personalized health monitoring and disease diagnosis (*Yang et al., 2021*). MN-based systems also possess strong technical abilities for photodynamic therapy, immunotherapy, and photothermal therapy (*Li et al., 2021*). MN stimulates the appropriate immune response by administering drugs close to underlying neutrophils, Langerhans, and dendritic cells found deep within the skin (*Singh & Kesharwani, 2021*). *Ma et al. (2015)* devised Poly (lactic-co-glycolic) acid (PLGA) nanoparticles enveloping Doxorubicin (DOX) and coated it on stainless steel MNs as an innovative approach for local and minimally invasive intratumoral drug delivery. DOX-PLGA-NP/MNs showed that DOX, when released from nanoparticles could dissipate up to four mm deep into porcine cadaver buccal tissues, and around 1–2 mm laterally from the point of insertion, causing cell death. Coated microneedles were found to perform better overall than hypodermic needles because they can deliver drugs at precise and evenly distributed locations in the tissue, as well as overcome drug loss and systemic side effects resulting from injection fluid leakage (*Ma et al., 2015*). Regarding therapeutic intervention of oral cancer. According to *Xu et al. (2021)* microneedles have the potential to be utilized for targeted delivery of anticancer medications to the specific site of oral cancer within the oral cavity. The use of this specific drug delivery strategy presents numerous benefits in comparison to conventional techniques of oral or systemic drug administration. These advantages encompass enhanced drug effectiveness, diminished adverse effects on surrounding tissues, and improved patient adherence (*Ganeson et al., 2023*). According to *Xu et al. (2021)* research has demonstrated that microneedles has the capability to be coated with drug-loaded nanoparticles or drug formulations, enabling the

attainment of regulated and sustained drug release specifically at the tumor site. According to *Ganeson et al. (2023)* the utilization of this approach enables accurate regulation of drug concentration by means of spatial and temporal manipulation of drug release, hence reducing the adverse impact on healthy tissues. Furthermore, the utilization of microneedles offers convenient application and offers a painless approach to medication delivery, hence serving as a patient-centric substitute for conventional methods of oral or injectable administration (*Qi et al., 2023*). In addition, previous studies have investigated the potential of microneedles in the administration of vaccines to the oral mucosa, a factor that may have implications in the field of oral cancer therapy (*Creighton & Woodrow, 2019*). According to *Ma et al. (2014)*, the utilization of coated microneedles presents a viable approach for administering various vaccine formulations to the tissues of the oral cavity, thereby stimulating both systemic and mucosal immune responses (*Creighton & Woodrow, 2019*; *Ma et al., 2014*). The aforementioned strategy has a notable benefit in terms of localized administration of vaccines to the tumor location, hence potentially augmenting the immune response directed towards malignant cells (*Creighton & Woodrow, 2019*). In general, microneedles have demonstrated significant promise in the therapeutic management of oral cancer by their ability to facilitate precise and regulated administration of pharmaceutical agents to the malignant area within the mouth cavity. This technique presents several benefits, including greater therapeutic efficacy, minimized side effects, and improved patient adherence. Additional investigation and advancement in this domain are necessary to enhance the design and composition of microneedles for the purpose of treating oral cancer, as well as to assess their effectiveness in clinical environments. Immunotherapies, namely immune checkpoint inhibitors (ICI) such as anti-PD-1 and anti-CTLA-4 antibodies, have emerged as a highly promising approach for the treatment of head and neck squamous cell carcinoma (HNSCC). These therapies aim to restore the body's natural immune response against tumor cells by targeting specific immune checkpoints. Despite their potential, it is important to note that the clinical benefit and long-lasting responses are observed in less than 20% of HNSCC patients. Furthermore, the practical use of immune checkpoint inhibitors (ICI) has been constrained due to immune-related adverse events (irAEs) that arise as a result of reduced peripheral immunological tolerance. While immune-related adverse events (irAEs) are typically reversible, they have the potential to escalate in severity, leading to premature discontinuation of therapy or posing a life-threatening risk. To mitigate the immune-related adverse events (irAEs) associated with systemic immune checkpoint inhibitor (ICI) therapy, Gilrdi et al. devised an innovative approach including the utilization of a soluble microneedle (MN) array for localized drug delivery. In this study, we utilized a syngeneic murine model of head and neck squamous cell carcinoma (HNSCC) that exhibits genetic similarities to tobacco induced HNSCC in humans. Through their investigation, they observed that the administration of anti-CTLA-4 therapy, either systemically or locally within the tumor microenvironment, resulted in a tumor response above 90%. This response was discovered to be reliant on the presence of CD8 T cells and a specific subset of dendritic cells known as conventional dendritic cell type 1 (cDC1) (*Gilardi et al., 2022*; *Gilardi et al., 2022*).

In summary, this study has provided an overview and synthesis of the present application of sophisticated drug delivery technologies, specifically quantum dots (QDs), carbon nanotubes (CNT), magnetic nanoparticles (MNPs), dendrimers, and microneedles (MNs), in the treatment of oral cancer. These delivery platforms had valuable diagnostic and therapeutic advantages over conventional methods or controls, such as improved tumor screening and visualization, improved tumor targeting and reduced toxicity to the adjacent non-cancerous tissues, improved drug's solubility and distribution hence maximizing its absorption and uptake into the tumor.

Despite the considerable potential exhibited by these delivery systems, they have been largely overlooked in the context of oral cancer, resulting in a limited number of research publications being published within the last decade. Furthermore, a significant number of research publications have not fully used the complete theranostics capabilities of these platforms. Instead, they have solely utilized these delivery systems for either diagnostic or therapeutic objectives. However, it is evident that modern drug delivery technologies are not being fully utilized in the treatment of oral cancer, highlighting the need for future research and exploration in this intriguing area. Nevertheless, the continuous progress in the field of nanomedicine and oncology has the promise of transforming the approach to oral cancer care. This entails the development of innovative systems that can potentially enhance treatment outcomes by delivering more precise, efficient, and patient-centric therapeutic interventions in the coming years.

## CONCLUSION

In conclusion, the results indicate that the utilization of sophisticated nano/micro delivery platforms exhibits potential in resolving a multitude of difficulties linked to chemotherapy. These platforms possess the capability to accurately target malignant cells, which presents an opportunity to minimize the negative impact on neighboring healthy tissues. Consequently, this advancement promotes the exploration of novel diagnostic and treatment approaches for oral cancer. Nevertheless, it is critical to acknowledge that the identification of the most effective drug delivery system for the management of oral cancer is contingent upon a multitude of factors, including the drug's particular attributes, the intended administration route, and the site of delivery. Additional investigation is required to determine the potential applications of other nano drug delivery systems, including dendrimers, magnetic nanoparticles, quantum dots, and microneedle drug delivery systems, in the context of treating oral cancer. Such systems would benefit from a more comprehensive understanding of their particular merits and drawbacks.

### Funding
The authors received no funding for this work.

### Competing Interests
The authors declare there are no competing interests.

## Author Contributions

- Asmaa Uthman conceived and designed the experiments, authored or reviewed drafts of the article, and approved the final draft.
- Noor AL-Rawi conceived and designed the experiments, performed the experiments, prepared figures and/or tables, and approved the final draft.
- Musab Hamed Saeed analyzed the data, authored or reviewed drafts of the article, and approved the final draft.
- Bassem Eid conceived and designed the experiments, analyzed the data, prepared figures and/or tables, and approved the final draft.
- Natheer H. Al-Rawi performed the experiments, authored or reviewed drafts of the article, and approved the final draft.

## Data Availability

This is a literature review.

## Supplemental Information

Supplemental information for this article can be found online at http://dx.doi.org/10.7717/peerj.16732#supplemental-information.

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
