# Peer review of "Tunable theranostics: innovative strategies in combating oral cancer"

_PeerJ, doi:10.7717/peerj.16732_

## Round 0.1 · original submission · Major Revisions

Reviewers have raised some concerns and shortcomings in the study. MAJOR revision is suggested, which requires substantial and thorough revision to appreciate the quality of the manuscript. Therefore, authors are requested to revise their manuscript in light of reviewers comments. Furthermore, reviewer has suggested that you cite specific references. You are welcome to add it/them, if you believe they are relevant. However, you are not required to include these citations, and if you do not include them, this will not influence my decision.

**Language Note:** The review process has identified that the English language must be improved. PeerJ can provide language editing services - please contact us at [email protected] for pricing (be sure to provide your manuscript number and title). Alternatively, you should make your own arrangements to improve the language quality and provide details in your response letter. – PeerJ Staff

Reviewer 1 ·

Basic reporting

Clear, unambiguous, professional English language has been used throughout the manuscript. Introduction and background are well explained and includes the context of the study. Literature is referenced well and are relevant to the article, but a few references are required to be added. I have mentioned about them later. The article structure conforms to PeerJ standards, and discipline norm, but can be improved for enrichment. The review is of broad and cross-disciplinary interest and within the scope of the journal. The field of research has not been reviewed recently. The Introduction adequately introduces the subject and make it clear who the audience is and what the motivation is.

Experimental design

Article content is within the Aims and Scope of the journal. Rigorous investigation was performed to a high technical & ethical standard. The Survey Methodology consistent with a comprehensive, unbiased coverage of the subject to a high extent, but a few references are needed to be included which I shall mention later. Yes, the sources are adequately cited, and the paraphrasing is good. The review has been organized logically into coherent paragraphs/subsections.

Validity of the findings

I have the following comments.
Only two original research (Ma et al. (2022) and Zhan g et al. (2020)) were cited and explained for section 1.Quantum Dots. More research should be explained using QDs or surface functionalized QDs for drug delivery for oral cancer.
In section 2. Carbon Nanotubes, a lot of introductions about carbon nanotubes is given which can be reduced. Regarding the application of carbon nanotubes only one work has been cited.
In Google scholar, as the authors have mentioned between 2012-2022 many articles are present and must be cited. Some are as follows:
1. Sánchez-Tirado E, Salvo C, González-Cortés A, Yáñez-Sedeño P, Langa F, Pingarrón JM. Electrochemical immunosensor for simultaneous determination of interleukin-1 beta and tumor necrosis factor alpha in serum and saliva using dual screen printed electrodes modified with functionalized double–walled carbon nanotubes. Analytica chimica acta. 2017 Mar 22;959:66-73.

In section 3. Magnetic Nanoparticles, a lot of introductions about magnetic nanoparticles is given. Regarding the application of carbon nanotubes only two works have been cited.
In Google scholar, as the authors have mentioned between 2012-2022 many articles are present and must be cited. Some are as follows:
1. Miao L, Liu C, Ge J, Yang W, Liu J, Sun W, Yang B, Zheng C, Sun H, Hu Q. Antitumor effect of TRAIL on oral squamous cell carcinoma using magnetic nanoparticle-mediated gene expression. Cell Biochemistry and Biophysics. 2014 Jul;69:663-72.
2. Candido NM, Calmon MD, Taboga SR, Bonilha JL, Santos SD, Lima EC, Sousa CR, Rahal P, Lacava ZG. High efficacy in hyperthermia-associated with polyphosphate magnetic nanoparticles for oral cancer treatment.
3. Tsai MT, Sun YS, Keerthi M, Panda AK, Dhawan U, Chang YH, Lai CF, Hsiao M, Wang HY, Chung RJ. Oral cancer theranostic application of feau bimetallic nanoparticles conjugated with MMP-1 antibody. Nanomaterials. 2021 Dec 27;12(1):61.

Additional comments

The conclusions are well stated, linked to original research question. There are well developed and supported argument that meets the goals set out in the Introduction. The Conclusion also identifies unresolved questions / gaps / future directions.
I recommend major revision.

·

Basic reporting

No comment

Experimental design

No comment

Validity of the findings

No comment

Additional comments

The article is nicely written and covers almost every component that falls under the theme of the article. But there are a few typos in the article including: as in table 1
....... whole body florescent images .........
suggestion should be fluorescent...

·

Basic reporting

1. The overall structure is not so clear and logical.
2. Additionally, the current review talked about too much basic information, but not prominent research progression. Therefore, the current version was not too helpful for readers.
3. Why you have not searched research papers on PubMed and other research sites?
4. Line 57-58, This is too old for this review. Provide data for 2023 or 2022.
5. Line 58-60, presently 2023 is running and in the last 10 years lots of significant improvement has been done in therapeutics. Update the recent data with recent citations.
6. Figures are not described clearly in the text.

Experimental design

1. Article content is within the aim and scope.
2. What are the basic criteria for the selection of quantum dots, carbon nanotubes, magnetic nanoparticles, dendrimers, and microneedles nano delivery systems?
3. However various articles show the potential role of carbon dots and polymeric NPs as a therapeutic drug delivery system in oral cancer.
4. The major importance of this article is the novel research being done and its results. In the text, these were vaguely described or completely skipped.
5. Table 1: Give specific limitations and advantages of nano/ micro formulations in oral cancer theragnostic.
6. In Table 2, the experimental periods and doses should be included, which are very important.
7. Table 2 is very short and does not cover all thermostatic approaches discussed in the article.

Validity of the findings

1. Without a more detailed explanation of current researchers, etc., and an expert opinion of the importance of the results, this review is mostly just information that could have been copied and pasted from other reviews already published.
2. According to your thorough study and review writing among all these systems any specific delivery system showing more potential or all are the same, discuss it.
3. What is the future significance of this study?

Additional comments

Research data is very poorly cited and insufficient.

---

## Round 0.2 · accepted · Accept

Manuscript is significantly improved by the authors and now can be accepted in its current form.

·

Basic reporting

1. All the suggested comments have been incorporated into the revised version.
2. The English language is professional and scientific terms have been used throughout the manuscript.
3. More recent and relevant references have been cited in the revised version.

Experimental design

1. Article content is within the Aims and Scope of the journal.
2. The methodology and design of the paper is logical and acceptable.

Validity of the findings

1. The result summarized in the revised table is sufficient to explain the overall finding of the work.

Additional comments

Paper can be acceptable in its present form.